# Microbial Communities and Metabolites of Whole Crop Corn Silage Inoculated with *Lentilactobacillus plantarum* and *Lentilactobacillus buchneri*

Qian Guo [1,†] , Xia Hao [1,†], Yuerui Li [1], Qing Zhang [2], Chao Wang [3,*] and Hongyan Han [1,*]

1 State Key Laboratory of Reproductive Regulation and Breeding of Grassland Livestock, School of Life Sciences, Inner Mongolia University, Hohhot 010070, China
2 Faculty of Forestry and Landscape Architecture, South China Agricultural University, Guangzhou 510642, China
3 Inner Mongolia Engineering Research Center of Development and Utilization of Microbial Resources in Silage, Inner Mongolia Academy of Agriculture and Animal Husbandry Science, Hohhot 010031, China
* Correspondence: wangchao@imaaahs.ac.cn (C.W.); hanhongyan1018@outlook.com (H.H.); Tel.: +86-471-529-8583 (H.H.)
† These authors contributed equally to this work.

**Abstract:** To investigate the effects of different types of lactic acid bacteria (LAB) on aerobic stability, microbial community and metabolites of whole crop corn silage ensiled with *Lentilactobacillus plantarum* (LP) and *Lentilactobacillus buchneri* (LB) or not (CK), the fermentation parameters, aerobic stability, microbial community and metabolite differential components of whole crop corn silage were analyzed after ensiling for 8 months. The results showed that the pH of the whole treatment was lower than 4.2, which indicates good fermentation quality. Compared with the LP group, the LB group significantly improved the aerobic stability of whole crop corn silage ($p < 0.05$). The addition of LB and LP both increased the number of LAB and the relative abundance of *Lentilactobacillus*. Metabolite analysis results showed that 28 metabolites were significantly different between the LP and CK groups ($p < 0.01$), 15 metabolites were significantly different between the LB and CK groups ($p < 0.01$), and 17 metabolites were significantly different between the LP and LB groups ($p < 0.01$). The antioxidant metabolites 9-oxo-10(E), 12(E)-octadecadienoic acid and 9(Z),11(E),13(E)-octadecatrienoic acid ethyl ester in the LB group were significantly higher than those in the lp group ($p < 0.01$). Therefore, compared with LP, obligate heterofermentative LB is more beneficial to maintain the stability of whole crop corn silage after cellar opening.

**Keywords:** *Lentilactobacillus plantarum*; *Lentilactobacillus buchneri*; aerobic stability; microbial community; metabolites

## 1. Introduction

Whole corn is a common source of roughage for ruminants in China. Whole corn has the characteristics of low buffering capacity and high soluble carbohydrates, with good silage fermentation quality [1]. In order to overcome the uneven seasonal distribution of feed resources, whole corn is mainly used to make whole crop corn silage. However, during the silage cellar-opening stage, once whole crop corn silage is exposed aerobically, often causing the rapid multiplication of molds and yeasts and leading to spoilage and deterioration of the silage and accumulation of toxins from undesirable microbial metabolism. This affects the health of ruminants during the feeding process and even leads to death [2,3]. Therefore, the key to ensure the quality of whole crop corn silage is to control the activities of undesirable microorganisms during the cellar-opening stage.

The lactic acid produced during ensiling reduces the pH value of silage [4]. *Lentilactobacillus plantarum* (LP) is a common facultative heterofermentative LAB that can reduce the contents of acetic acid, butyric acid and ammonia nitrogen in silage. It can also reduce

the pH of silage and increase the content of lactic acid and dry matter (DM) recovery. *Lentilactobacillus buchneri* (LB), an obligate heterofermentative LAB, has been successfully used as an additive to improve the aerobic stability of corn silage, and can convert lactic acid into acetic acid and 1,2-propanediol [5]. Acetic acid has antifungal properties that can inhibit the reproduction of undesirable microorganisms such as yeast and achieve the effect of improving the aerobic stability of silage [6]. In addition to improving aerobic stability, some LB can produce ferulic acid (esterase) in silage, which has the potential to improve fiber digestibility [7]. The addition of LP and LB to whole crop corn silage is an effective way to control the activities of undesirable microorganisms. LAB used in silage inoculants can survive in rumen fluids, interact with rumen microorganisms, change rumen fermentation, enhance rumen function, and provide probiotic effects in the small intestine [8]. Feeding silage inoculated with LAB additive can improve animal feed intake and production performance to varying degrees. Unfortunately, the function of LAB additive is unstable, and the mechanism of action is still unclear [6].

This study explored the effects of LP and LB on fermentation quality, aerobic stability, and metabolites of whole crop corn silage. In order to facilitate transportation, the whole crop corn was used for bale silage in practical production. Therefore, this study prepared 30 kg of bale silage to evaluate the effects of LP and LB on the fermentation quality, aerobic stability, microbial composition and metabolites of corn silage, aiming to reveal the inoculation effect and mechanism of these two strains.

## 2. Materials and Methods

### 2.1. Raw Materials and Experimental Design

The whole corn plant was selected around Ulanqab City, Inner Mongolia, China at half milk line stage with DM content of 38.9%. The whole corn was cut into 1–2 cm long stalks immediately after mowing. LP and LB bacteria powders were added to 30 kg of raw materials with a final addition amount of $10^5$ cfu/g and sprayed with 300 g water to activate the strain. The whole corn with the same amount of water was used as the control group (CK). The whole corn was mixed well and put into a 50 kg plastic bag and vacuumed with a household vacuum cleaner. Use the same time to vacuumed it. After vacuuming, the outer layer of whole crop corn silage was covered with a fiber bag to prevent damage during movement. Then, 300 kg of whole crop corn silage was prepared for each treatment (10 bags per group, 30 bags in total) and stored in the feed shed for 8 months. After opening, 3 bags from 10 bags for each treatment were emptied, fully mixed, and then evenly sampled from multiple points. From each bag, 2 kg was taken for the determination of fermentation quality, aerobic stability, microbial community, and metabolites.

### 2.2. Silage Quality and Aerobic Stability Analysis

2.2.1. Fermentation Quality Analysis

Whole 2 kg corn silage samples were mixed evenly. Then, 20 g was taken using the five-point sampling method and mixed with 180 mL distilled water, placed in a refrigerator at 4 °C for 24 h, coarsely filtered with 4 layers of sterile gauze, and finally filtered with medium-speed qualitative filter paper to obtain the extract. The pH of the extract was determined using a pH meter (PB-10, sartorius, Göttingen, Germany) as per He et al. [9], and the content of organic acids, lactic acid, acetic acid, propionic acid and butyric acid was determined by high-performance liquid chromatography.

2.2.2. Microbial Population Analysis

Another 20 g of whole crop corn silage sample was taken and mixed with 180 mL of sterile saline (0.85% NaCl, *w/v*), and after standing for 30 min, the bacterial suspension was diluted from $10^{-1}$ times equal gradient to $10^{-6}$ times. LAB was incubated with MRS agar medium in a constant-temperature incubator at 30 °C for 48 h, *Escherichia coli* was incubated with violet red bile agar in the incubator at 30 °C for 24 h and yeast was incubated with

potato dextrose agar in the incubator at 28 °C for 48 h. All microorganisms were determined by plate counting, and the microbial population was counted by lg cfu/g fresh matter (FM).

### 2.2.3. Aerobic Stability Analysis

About 150 g of corn silage was taken and put into a 500 mL plastic bottle and a continuous thermometer probe inserted to record the temperature every 15 min for 7 consecutive days. The internal temperature of the sample 2 °C above room temperature was taken as a sign of the beginning of spoilage to judge aerobic stability.

### 2.3. Bacterial and Microbial Community Analysis

The bacterial DNA in silage was extracted using a soil DNA kit (Guangzhou Danmai Biotechnology, Guangzhou, China) as per Chen et al. [10]. The 16S rDNA V3–V4 region of bacteria was amplified by polymerase chain reaction. A DNA gel recovery kit (Axygen Biosciences, Union City, CA, USA) was used to recover amplicons according to the instructions. Then, an ABI StepOnePlus real-time PCR system was used for quantitative analysis for amplicons, and an Illumina platform was used to sequence the purified amplicons. The original sequence was assembled and analyzed as per Liu et al. [11]. All original data have been uploaded to NCBI under the login number PRJNA61214.

### 2.4. Metabolite Analysis

The silage samples were extracted with 50% methanol buffer. Briefly, 20 μL of sample was extracted with 120 μL of precooled 50% methanol, vortexed for 1 min, and then incubated at room temperature for 10 min. The extraction mixture was then stored overnight at −20 °C. After centrifugation at 4000 g for 20 min and filtration through a 0.22 μm microspore filter membrane, the supernatants were analyzed using an UPLC-ESI-MS/MS system as per Hu et al. (2020) [10].

### 2.5. Statistical Analysis

SPSS 19.0 was used to conduct variance analysis on the fermentation quality and aerobic stability of whole crop corn silage. Univariate analysis of the general linear model was carried out for significance of difference analysis, and Duncan multiple comparative analysis was carried out between different treatments. The significance of means was tested using Duncan's multiple range test, with $p < 0.05$ indicating significant difference. A Shapiro–Wilk test value of $p > 0.05$ indicates a normal distribution. Levene's test was used to assess homogeneity of variance, with $p > 0.05$ indicating homogeneity.

## 3. Results

### 3.1. Effects of LAB on the Fermentation Quality and Aerobic Stability of Whole Crop Corn Silage

The effects of LAB on the fermentation quality and aerobic stability of whole crop corn silage are shown in Table 1 [12]. LAB had a very significant effect on DM content ($p < 0.01$) and a significant effect on yeast quantity ($p < 0.05$). Compared with CK and LB, LP significantly increased the DM content of silage ($p < 0.05$). The pH values were all lower than 4.0, and there was no significant difference between the two treatment groups and CK. Compared with CK, LP significantly increased the amount of yeast ($p < 0.05$), while LB had no significant effect. Lactic acid and acetic acid content in the LP group was lower than that in the CK and LB groups, the content of lactic acid in the CK group was the highest, and the content of acetic acid in the LP group was the highest. Compared with the LP group, the CK and LB groups showed stronger aerobic stability, but there was no significant difference.

**Table 1.** Fermentation quality and aerobic stability of whole crop corn silage [12].

| Item | CK | LP | LB | SEM | *p* Value |
|---|---|---|---|---|---|
| DM (%) | 32.1 b | 37.0 a | 32.9 b | 0.80 | 0.001 |
| pH | 3.89 | 3.93 | 3.86 | 0.01 | 0.089 |
| *Lactic acid bacteria* (log cfu/g) | 3.60 | 5.28 | 4.11 | 0.35 | 0.128 |
| *Coliform bacteria* (log cfu/g) | <2.00 | <2.00 | <2.00 | <0.01 | - |
| *Yeasts* ($\log_{10}$ cfu/g FM) | <2.00 b | 4.24 a | 2.60 ab | 0.73 | 0.023 |
| Lactic acid (%DM) | 5.55 | 3.64 | 5.39 | 0.99 | 0.737 |
| Acetic acid (%DM) | 2.00 | 1.45 | 2.28 | 0.44 | 0.788 |
| Aerobic Stability (h) | 68 | 36 | 128 | 17.9 | 0.081 |

DM, dry matter content; CK, control silage; LP, corn silage treated with *Lentilactobacillus plantarum*; LB, corn silage treated with *Lentilactobacillus buchneri*; SEM, standard error of mean. Values with different letters (a, b) show significant differences among treatments.

### 3.2. α-Diversity Analysis of Bacterial Community of Whole Crop Corn Silage

The α-diversity index of the effect of LAB on the bacterial community of whole crop corn silage is shown in Table 2. The Shannon index and Simpson index representing microbial diversity and homogeneity were significantly different between whole crop corn silage and whole corn stover with different treatments ($p < 0.01$), but there was no significant difference between OTUs and Chao1. Among all indices, the value of the corn stover was the lowest, the value of the CK group was the highest, and the value of the LP and LB groups were close.

**Table 2.** α-Diversity index of bacterial community in whole crop corn silage.

| Item | OTUs | Shannon | Simpson | Chao1 | Good's Coverage |
|---|---|---|---|---|---|
| CS | 183 | 3.58 | 0.82 | 195 | 1.00 |
| CK | 350 | 5.09 | 0.93 | 359 | 1.00 |
| LP | 257 | 4.42 | 0.89 | 264 | 1.00 |
| LB | 259 | 4.52 | 0.90 | 269 | 1.00 |
| SEM | 24.8 | 0.17 | 0.01 | 29.1 | - |
| *p* value | 0.182 | 0.000 | 0.000 | 0.275 | - |

CS, corn stover; CK, control silage; LP, corn silage treated with *Lentilactobacillus plantarum*; LB, corn silage treated with *Lentilactobacillus buchneri*; SEM, standard error of mean.

### 3.3. Effects of LAB on the Bacterial Community of Whole Crop Corn Silage

The relative abundance of dominant bacterial phyla in the bacterial community in whole crop corn silage is shown in Table 3. *Firmicutes* occupied a dominant position in all samples, followed by *Proteobacteria* and *Cyanobacteria*. The relative abundance of *Firmicutes* and *Cyanobacteria* in each treatment group was significantly different ($p < 0.01$), and the difference of *Proteobacteria* was significant ($p < 0.05$). The relative abundance of *Firmicutes* in corn stover was 94.1%, CK 43.9%, in LP 53.9%, and LB 60.1%. After ensiling, the relative abundance of *Proteobacteria* and *Cyanobacteria* in each treatment group significantly increased. The relative abundance of *Proteobacteria* in CK, LP and LB was 32.3%, 25.4% and 17.0%, respectively, and that of *Cyanobacteria* was 18.3% and 18.43%, and 19.9%, respectively, much higher than those in corn stover, which were 5.03% and 0, respectively.

The relative abundance of dominant bacterial phyla in the bacterial community in whole crop corn silage is shown in Figure 1 and Table 4. In corn stover, the relative abundance of *Bacillus* was 89.0%, occupying an absolute dominant position. After ensiling, the bacterial community in whole crop corn silage changed significantly. The dominant bacterial genera—*Lentilactobacillus*, *Bacillus*, and *Pediococcus*—showed extremely significant differences among the groups ($p < 0.01$), and *Leuconostoc* and *Sphingolipids* showed significant differences ($p < 0.05$). After ensiling, *Lentilactobacillus* occupied the dominant position in the bacterial community. In the CK, LP, and LB groups, the relative abundance of *Lentilactobacillus* was 27.4%, 42.8 and 49.6%, respectively. Compared with the CK group,

LP and LB additives increased the relative abundance of *Lentilactobacillus* and decreased the relative abundance of *Leuconostoc.*

**Table 3.** Relative abundance of dominant bacterial phyla in whole crop corn silage.

| Phylum | CS | CK | LP | LB | SEM | *p*-Value |
|---|---|---|---|---|---|---|
| *Firmicutes* | 94.1 | 43.9 | 53.9 | 60.1 | 5.92 | 0.000 |
| *Proteobacteria* | 5.03 | 32.3 | 25.4 | 17.0 | 3.58 | 0.011 |
| *Cyanobacteria* | 0.00 | 18.3 | 18.43 | 19.9 | 2.64 | 0.001 |
| *Bacteroidetes* | 0.10 | 1.37 | 0.39 | 0.53 | 0.15 | 0.001 |
| *Actinobacteria* | 0.02 | 0.42 | 0.31 | 0.36 | 0.05 | 0.006 |
| *Deinococcus-Thermus* | 0.00 | 0.36 | 0.17 | 0.54 | 0.09 | 0.127 |
| *Acidobacteria* | 0.04 | 0.48 | 0.31 | 0.09 | 0.08 | 0.184 |
| *Planctomycetes* | 0.00 | 0.10 | 0.03 | 0.02 | 0.01 | 0.001 |
| *Verrucomicrobia* | 0.00 | 0.10 | 0.01 | 0.02 | 0.01 | 0.016 |
| *Unclassified* | 2.49 | 0.95 | 1.36 | 0.74 | 0.25 | 0.024 |

CS, corn stover; CK, control silage; LP, corn silage treated with *Lentilactobacillus plantarum*; LB, corn silage treated with *Lentilactobacillus buchneri*; SEM, standard error of mean.

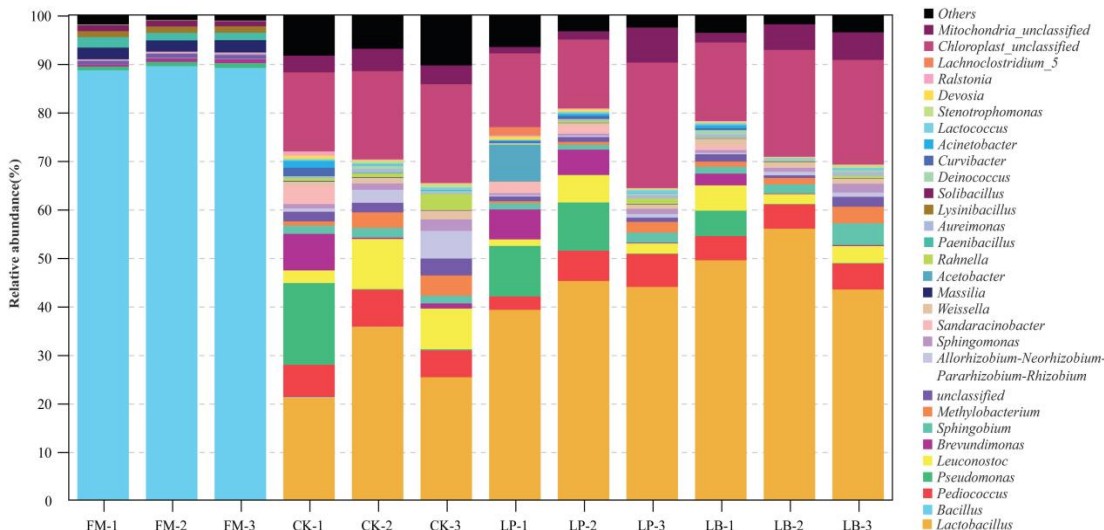

**Figure 1.** Relative abundance of bacterial community in whole crop corn silage at genus level. Note: FM is corn stover. CK, control silage; LP, corn silage treated with *Lentilactobacillus plantarum*; LB, corn silage treated with *Lentilactobacillus buchneri*.

**Table 4.** Relative abundance of dominant bacterial genera in whole crop corn silage.

| Genus | CS | CK | LP | LB | SEM | *p*-Value |
|---|---|---|---|---|---|---|
| *Lactobacillus* | 0.04 | 27.4 | 42.8 | 49.6 | 5.88 | 0.000 |
| *Bacillus* | 89.0 | 0.05 | 0.00 | 0.00 | 11.62 | 0.000 |
| *Pediococcus* | 0.00 | 6.61 | 5.27 | 5.12 | 0.82 | 0.001 |
| *Pseudomonas* | 0.86 | 5.74 | 6.81 | 1.82 | 1.62 | 0.552 |
| *Leuconostoc* | 0.00 | 7.11 | 3.04 | 3.57 | 0.97 | 0.046 |
| *Brevundimonas* | 0.67 | 2.97 | 3.85 | 0.93 | 0.77 | 0.430 |
| *Sphingobium* | 0.07 | 1.75 | 1.45 | 2.57 | 0.35 | 0.046 |

CS, corn stover; CK, control silage; LP, corn silage treated with *Lentilactobacillus plantarum*; LB, corn silage treated with *Lentilactobacillus buchneri*; SEM, standard error of mean.

### 3.4. Metabolite Analysis of Whole Crop Corn Silage

Principal component analysis (PCA) and differential volcano plots of metabolites in whole crop corn silage with different treatments are shown in Figure 2. PCA showed that there were certain metabolite differences between CK and corn stover, CK and LP, and CK and LB (Figure 2A–C). The differential volcano plots showed that there were

a considerable number of metabolite differences between the above treatment groups (Figure 2a–c). Among them, the metabolites between the CK group and corn stover had the most obvious difference, with the largest fold difference, and 222 metabolites had extremely significant differences ($p < 0.01$). The fold difference between the CK group and the LP group was the second largest, and 28 metabolites were extremely significantly different ($p < 0.01$). The fold difference was smallest between the CK group and the LB group, and 15 metabolites were extremely significantly different ($p < 0.01$). Seventeen metabolites were extremely significantly different between the LB group and the LP group ($p < 0.01$) (Table 5).

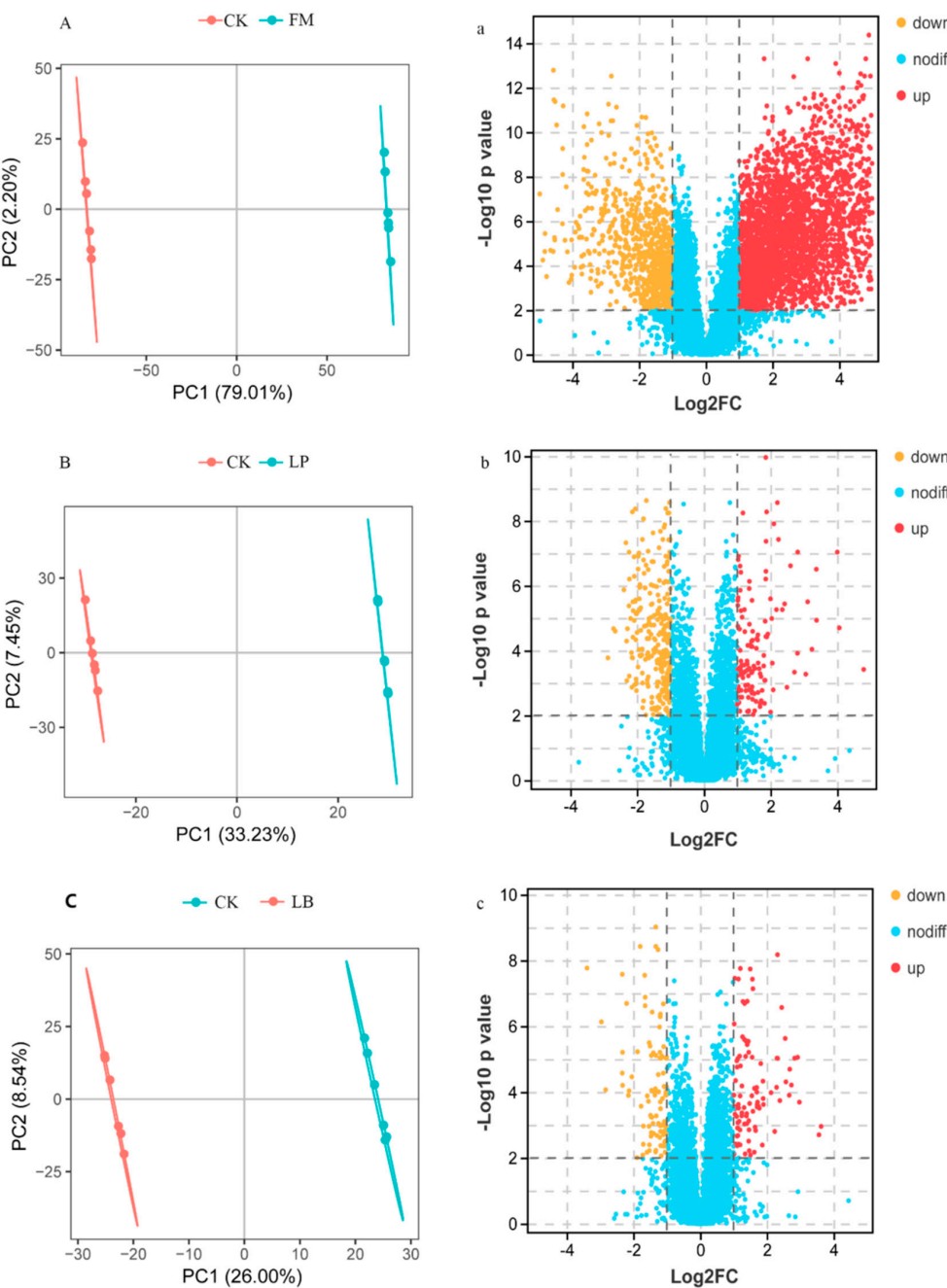

**Figure 2.** Differential volcano plots of metabolite differences of whole crop corn silage. Note: (**A**–**C**) is principal component analysis of metabolites in whole crop corn silage with different treatments. (**a**–**c**) is differential volcano plots of metabolites in whole crop corn silage with different treatments. FM is corn stover. CK, control silage; LP, corn silage treated with *Lentilactobacillus plantarum*; LB, corn silage treated with *Lentilactobacillus buchneri*.

**Table 5.** Metabolite differences of whole crop corn silage between LB and LP groups.

| Metabolite | FC (LB/LP) | Important Predictor Variables | *p*-Value |
|---|---|---|---|
| Gly-Val | 3.32 | 4.20 | 0.00 |
| 9-Oxo-10(E),12(E)-octadecadienoic acid | 3.22 | 3.11 | 0.00 |
| 9Z,11E,13E-Octadecatrienoic acid ethyl ester | 2.92 | 2.97 | 0.00 |
| 5Z,11Z,14Z-Eicosatrienoic acid | 2.86 | 2.79 | 0.00 |
| 8(9)-Epoxy-5Z,11Z,14Z-eicosatrienoic acid | 2.69 | 3.75 | 0.00 |
| 8,11-Tridecadienoic acid | 2.58 | 3.76 | 0.00 |
| Diniconazole | 2.40 | 3.02 | 0.00 |
| Testosterone_Propionate | 2.35 | 2.41 | 0.00 |
| Acylcarnitine 15:0 | 2.30 | 3.11 | 0.00 |
| Bis(2-ethylhexyl) adipate | 2.25 | 2.97 | 0.00 |
| Homoorientin | 2.10 | 2.52 | 0.00 |
| Quercetin-3-O-rhamnoside | 2.04 | 3.15 | 0.00 |
| LysoPC 16:0 | 0.49 | 2.32 | 0.00 |
| Acylcarnitine 25:4 | 0.45 | 3.17 | 0.00 |
| LysoPC 18:2 | 0.44 | 2.61 | 0.00 |
| Pheophorbide a | 0.42 | 2.86 | 0.01 |
| Butanal | 0.36 | 3.11 | 0.00 |

LP, corn silage treated with *Lentilactobacillus plantarum*; LB, corn silage treated with *Lentilactobacillus buchneri*.

The significantly different metabolites between the LP group and CK group, LB group and CK group, and LB group and LP group after aerobic exposure are shown in Tables 5–7, with a total of 60 different metabolites between the groups. Compared with the CK group, seven metabolites in the LP group were significantly higher ($p < 0.01$), such as soyasaponin Bb, D-pantothenic acid and LysoPC 16:0, and 21 metabolites were significantly lower ($p < 0.01$), such as gerberinol, acylcarnitine 15:0 and LysoPC 18:2 (Table 5). Compared with the CK group, 10 metabolites in the LB group were significantly higher ($p < 0.01$), such as epsilon-caprolactam, hordenine and D-pantothenic acid, and five metabolites were significantly lower ($p < 0.01$), such as pheophorbide a, acylcarnitine 25:4 and LysoPC 16:0 (Table 6). Compared with the LP group, 12 metabolites in the LB group were significantly higher ($p < 0.01$), such as Gly Val, 9-oxo-10(E), 12(E)-octadecadienoic acid and 9(Z),11(E),13(E)-octadecatrienoic acid ethyl ester, and five metabolites were significantly lower ($p < 0.01$), such as butanal, pheophorbide a and LysoPC 18:2 (Table 6).

**Table 6.** Metabolite differences of whole crop corn silage between LP and CK groups.

| Metabolite | FC (LP/CK) | Important Predictor Variables | *p*-Value |
|---|---|---|---|
| Soyasaponin Bb | 6.05 | 4.34 | 0.00 |
| D-Pantothenic acid | 4.71 | 3.86 | 0.00 |
| LysoPC 16:0 | 3.58 | 3.74 | 0.00 |
| LysoPE 16:0 | 2.83 | 2.54 | 0.00 |
| Ser-Phe | 2.67 | 2.73 | 0.00 |
| 4-Hydroxy-7-trifluoromethyl-3-quinolinecarboxylic acid | 2.25 | 1.72 | 0.00 |
| Phytosphingosine | 0.50 | 1.72 | 0.00 |
| 8(9)-Epoxy-5Z,11Z,14Z-eicosatrienoic acid | 0.49 | 2.56 | 0.00 |
| 7-O-Acetylaustroinulin | 0.47 | 1.56 | 0.00 |
| 9-HODE | 0.46 | 1.54 | 0.00 |
| 12,13-Dihydroxy-9Z-octadecenoic acid | 0.46 | 1.66 | 0.00 |
| 2-Hydroxy-3-methoxybenzaldehyde | 0.45 | 1.42 | 0.00 |
| D-erythro-N-stearoylsphingosine | 0.44 | 2.46 | 0.00 |
| Methionine | 0.44 | 1.41 | 0.00 |
| 8Z,14Z-Eicosadienoic acid | 0.43 | 2.30 | 0.00 |

**Table 6.** *Cont.*

| Metabolite | FC (LP/CK) | Important Predictor Variables | *p*-Value |
|---|---|---|---|
| 5Z,11Z,14Z-Eicosatrienoic acid | 0.43 | 2.12 | 0.00 |
| Testosterone_Propionate | 0.40 | 2.66 | 0.00 |
| Isoquercitrin | 0.39 | 2.39 | 0.01 |
| Bis(2-ethylhexyl) adipate | 0.37 | 2.65 | 0.00 |
| (S)-2,3,4,5-tetrahydropyridine-2-carboxylate | 0.35 | 3.09 | 0.00 |
| Acylcarnitine 25:4 | 0.35 | 3.46 | 0.00 |
| epsilon-Caprolactam | 0.33 | 2.88 | 0.00 |
| Quercetin 3-galactoside | 0.32 | 2.47 | 0.00 |
| 8,11-Tridecadienoic acid, 13-(3-pentyl-2-oxiranyl)-, (8Z,11Z)- | 0.28 | 3.71 | 0.00 |
| LysoPC 18:2 | 0.26 | 3.84 | 0.00 |
| Acylcarnitine 15:0 | 0.25 | 2.87 | 0.00 |
| Gerberinol | 0.23 | 2.96 | 0.00 |

CK, control silage; LP, corn silage treated with *Lentilactobacillus plantarum*.

**Table 7.** Metabolite differences of whole crop corn silage between LB and CK groups.

| Metabolite | FC (LB/CK) | Important Predictor Variables | *p*-Value |
|---|---|---|---|
| Epsilon-caprolactam | 2.74 | 3.04 | 0.00 |
| Hordenine | 2.65 | 2.86 | 0.00 |
| D-Pantothenic acid | 2.63 | 3.41 | 0.00 |
| Phytosphingosine | 2.48 | 3.69 | 0.00 |
| Diniconazole | 2.31 | 3.39 | 0.00 |
| Karakoline | 2.22 | 2.89 | 0.00 |
| Diosgenin | 2.18 | 2.51 | 0.00 |
| Homoorientin | 2.06 | 2.45 | 0.00 |
| Quercetin-3-O-rhamnoside | 2.06 | 1.75 | 0.00 |
| LysoPE 16:0 | 2.03 | 2.52 | 0.00 |
| Isoquercitrin | 0.47 | 2.98 | 0.00 |
| Quercetin 3-galactoside | 0.35 | 3.55 | 0.00 |
| LysoPC 16:0 | 0.32 | 3.94 | 0.00 |
| Acylcarnitine 25:4 | 0.32 | 4.05 | 0.00 |
| Pheophorbide a | 0.14 | 5.74 | 0.00 |

CK, control silage; LB, corn silage treated with *Lentilactobacillus buchneri*.

## 4. Discussion

### 4.1. Effects of LAB on the Fermentation Quality and Aerobic Stability of Whole Crop Corn Silage

Ensiling is a process in which LAB convert soluble carbohydrates into organic acids under anaerobic conditions, thereby reducing the pH of silage environment and inhibiting undesirable microorganisms [13]. pH is the most intuitive index for evaluating the fermentation quality of silage, and pH lower than 4.2 indicates good silage fermentation [14]. In this study, the pH values of all treatment groups were lower than 4.2, indicating good fermentation quality of the whole corn. The pH value of the LP group was the highest, and the contents of lactic acid and acetic acid of LP group were the lowest among all treatment groups. In addition, the population of dominant microorganisms also reflects the differences of silage fermentation quality to a certain extent. Compared with CK group, the addition of LB and LP both increased the number of LAB in whole crop corn silage. Similarly, Wei et al. [15] reported that adding LP to corn silage increased the number of LAB. Zhao et al. [16] treated the whole corn with LB or LP, and found that the number of LAB increased after ensiling for 240 days. Schmidt et al. [17] and Mari et al. [18] also reported that the number of LAB was higher in corn silage treated with *Lentilactobacillus buchneri* than that in untreated corn silage, and further confirmed that the number of *Lentilactobacillus buchneri* in the treated silage was significantly higher. Yeasts are facul-

tative anaerobes that are acid tolerant and rely on lactic acid and soluble carbohydrates for reproduction even in the environment with pH lower than 3.5, resulting in lower nutritional quality of silage [19,20]. In this study, the number of yeast in the lp group was 4.24 lg cfu/g FM, which was much higher than that in the lb group (2.6 lg cfu/g FM) and CK group (2.00 lg cfu/g FM), probably due to the increase of available growth substrate (lactic acid accumulation produced by homofermentative LAB) and the decrease of antifungal component in silage (decrease of acetic acid concentration). The metabolism of lactic acid by yeast will eventually lead to the decrease of lactic acid content and the increase of the pH value of silage. When the pH value rises to 4.5, a large number of aerobic spoilage microorganisms will multiply, which in turn will cause spoilage and deterioration of silage and reduce the aerobic stability of silage. Therefore, the higher number of yeasts in the LP group was part of the reason why the aerobic stability of the LP group was lower than that of the CK and LB groups. In addition to the metabolic activities of microorganisms that affect the aerobic stability of silage, some metabolites also affect the aerobic stability of silage. Acetic acid has strong antifungal properties, and increased acetic acid will lead to improvement in aerobic stability. Danner et al. [21] showed that aerobic stability is highly dependent on acetic acid content. The addition of homofermentative LAB reduced aerobic stability, and the addition of *Lentilactobacillus buchneri* improved aerobic stability. This experiment also confirmed this result. The increase of acetic acid content in the LB group was closely related to the slow conversion of lactic acid into acetic acid by LB. The research results of Ranjit et al. [22], Nishino et al. [23] and Liu et al. [24] are consistent with this experiment, i.e., the addition of LB to silage can increase the acetic acid content and effectively improve the aerobic stability of silage. Muck et al. [9] noted that in about a third of studies, homofermentative inoculants reduced aerobic stability and that this situation occurred more often in corn silage than in grass or legume silage. Decreased acetic acid concentration after inoculation of homofermentative LAB may also lead to acceleration of yeast growth, thereby reducing aerobic stability. Overall, more research is needed to understand why the results of inoculation of homofermentative LAB are inconsistent among studies.

### 4.2. Effects of LAB on the Bacterial Community of Whole Crop Corn Silage

Silage quality is closely related to the evolution of bacterial communities. In the α-diversity analysis, the Shannon index and Simpson index decreased significantly with the addition of LB and LP, indicating that the population of bacterial communities in the LB and LP groups decreased, and some bacterial communities occupied a dominant position. Compared with the CK group, the relative abundance of *Firmicutes* at the bacterial community phyla level in the LB and LP groups increased significantly, the relative abundance of *Proteobacteria* decreased significantly (Table 3), and the relative abundance of *Lentilactobacillus* at the bacterial community genus level increased significantly (Table 4). This further explains the change process of the bacterial community during the fermentation process of the whole crop corn. It can be seen from Figure 1 that *Lentilactobacillus* played a dominant role in whole crop corn silage, and the addition of LB and LP further increased the relative abundance of *Lentilactobacillus* in silage. Similarly, Bai et al. [25] reported that the addition of LP to alfalfa can increase the relative abundance of *Lentilactobacillus*, improve the fermentation quality of silage, and improve the aerobic stability of alfalfa. Sun et al. [26] reported that the addition of LB to mulberry silage can increase the relative bacterial abundance of *Lentilactobacillus*. *Leuconostoc* is a heterofermentative LAB. Its metabolites are lactic acid, acetic acid, ethanol, etc., which can promote the fermentation of silage, but due to its poor acid resistance, its relative abundance is usually low after fermentation [27].

### 4.3. Effects of LAB on the Metabolites of Whole Crop Corn Silage

Metabolite analysis of whole crop corn silage showed that the inoculation of homofermentative LAB LP and heterofermentative LAB LB had different regulatory effects on the metabolite composition during the silage process. Soyasaponin Bb is a product secreted by soybean roots in soil, and some microorganisms such as Sphingomonadaceae and Caulobac-

teraceae are able to use soyasaponin Bb for enrichment, while soyasaponin Bb has a certain inhibitory effect on Enterobacteriaceae [28]. D-pantothenic acid, an essential vitamin widely used in medicine, food, and animal feed, plays an important role in protecting brain neurons [29,30]. Therefore, the higher content of soyasaponin Bb and D-pantothenic acid in the LB and LP groups improved the nutritional quality and health effects of the whole corn to some extent. 9-oxo-10(E), 12(E)-octadecadienoic acid is a derivative of conjugated linoleic acid, which has efficacy against cancer, obesity, atherosclerosis, and diabetes [31]. 9(Z),11(E),13(E)-octadecatrienoic acid is a derivative of α-linolenic acid and has the same physiological effect. Its antioxidant capacity is greater than that of α-linolenic acid, which can form physiologically active EPA and DHA in the human body. Long-term consumption of products containing α-linolenic acid or its derivatives can inhibit allergies and prevent cardiovascular and cerebrovascular diseases. Therefore, the contents of 9-oxo-10(E), 12(E)-octadecadienoic acid and 9(Z),11(E),13(E)-octadecatrienoic acid in LB are higher than those in LP, which may be the reason that LB has longer aerobic stability.

## 5. Conclusions

LB significantly improved the aerobic stability of whole crop corn silage, while LP decreased the aerobic stability of whole crop corn silage, and the aerobic stability of the LB group was significantly higher than that of the LP group ($p < 0.05$). Both LB and LP increased the number of LAB and the relative abundance of *Lentilactobacillus*. In sum, 28 metabolites were significantly different between the LP and CK groups, 15 metabolites were significantly different between the LB and CK groups, and 17 metabolites were significantly different between the LP and LB groups.

**Author Contributions:** Writing and data curation, Q.G.; investigation and data curation, Q.G. and X.H.; supervision, H.H. and C.W.; writing—review and editing, Q.G., X.H. and Q.Z.; project administration, funding acquisition, and writing—review and editing, H.H. and C.W.; formal analysis and methodology, Y.L. and Q.Z. All authors have read and agreed to the published version of the manuscript.

**Funding:** This study was funded by the National Natural Science Foundation of China (NSFC) (31902192) and Science and the Technology Major Project of Inner Mongolia Autonomous Region of China (zdzx2018065).

**Institutional Review Board Statement:** Not applicable.

**Informed Consent Statement:** Not applicable.

**Data Availability Statement:** All 16S rRNA sequences were submitted to the National Center for Biotechnology Information (NCBI), temporary submission ID: SUB7118893.

**Conflicts of Interest:** The authors declare that they have no conflict of interest.

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
