# Peer review of "Microbial Communities and Metabolites of Whole Crop Corn Silage Inoculated with Lentilactobacillus plantarum and Lentilactobacillus buchneri"

_processes, doi:10.3390/pr10112369_

Round 1
Reviewer 1 Report
I have reviewed the manuscript and determined that the following issue must be addressed before it can be considered for publication.
- How this study is relevant from practical point of view?
- The name of microorganisms should be italic in whole text of manuscript, especially in the title.
- Please define in the text the meaning of each acronym or abbreviation before starting to use it.
- What is the novelty of this research? The authors should present a clearer and concise presentation to highlight the novelty and significance of this paper.
- Materials section can be enhanced with real time images.
- The number of replication is not clear.
- Line 92, size distribution of crop corn silage sample is not clear.
- Line 101, 10-1 and 10-6, -1 & -6 should be superscript.
- Line 102, Escherichia coli should be in italic format.
- Line 106, what is FM?
- Although statistical analysis appears to have been performed, there is no text in the methods section providing the details of the analysis. Please add this text. In addition to including the details about ANOVA and the post hoc tests used, you must also present the details about the tests of the assumptions of ANOVS (normality and homogeneity of variance).
Reviewer 2 Report
The manuscript was well written and the authors have presented a study on the aerobic stabilty of silage with the potetial impact of lactic acid bacteria types. The study is topical as it has significant implication on cow health and the utilisation of ensiled materials for feeding.
The major comment is the ommission of key details in the methodology as well as minor suggestions detailed below:
Introduction
Page 2 Paragraph beginning with Lactic acid bacteria… correct sentences. “The lactic acid produced during ensiling reduces the pH value of silage thereby inhibiting the growth…
Last paragraph: The first sentence is unclear. The paragraph should highlight what research gap has been identified and the concluding sentence which describes the aim of the current study can well fit it.
Materials and Method
What quantity of water was used in the mixing? This will indicate the DM and moisture level of the materials before filling them into silo bags.
Ensiling process: Any indication of the density of the silo when filled? Volume vs weight? That is where compaction comes in. Adequate compaction complemented with a vacuum would have ensured more oxygen is eliminated.
How were the samples obtained from each bag? Describe the method in detail.
Discussion
Sentence correction. Highly dependent on the acetic acid content
Conclusions
Delete the first sentence.
Round 2
Reviewer 1 Report
The work and written of the manuscript is good.